# Ground-state and thermodynamic properties of the spin-$\frac{1}{2}$ Heisenberg model on the anisotropic triangular lattice

**Matías G. Gonzalez[1*], Bernard Bernu[1], Laurent Pierre[2] and Laura Messio[1,3]**

**1** Sorbonne Université, CNRS, Laboratoire de Physique Théorique
de la Matière Condensée, LPTMC, F-75005 Paris, France
**2** Université Paris X, UFR SGMI (Sciences gestion, mathématique et informatique),
F-92000 Nanterre, France
**3** Institut Universitaire de France (IUF), F-75005 Paris, France

⋆ matias.gonzalez@lptmc.jussieu.fr

## Abstract

The spin-$\frac{1}{2}$ triangular lattice antiferromagnetic Heisenberg model has been for a long time the prototypical model of magnetic frustration. However, only very recently this model has been proposed to be realized in the $Ba_8CoNb_6O_{24}$ compound. The ground-state and thermodynamic properties are evaluated from a high-temperature series expansion interpolation method, called "entropy method", and compared to experiments. We find a ground-state energy $e_0 = -0.5445(2)$ in perfect agreement with exact diagonalization results. We also calculate the specific heat and entropy at all temperatures, finding a good agreement with the latest experiments, and evaluate which further interactions could improve the comparison. We explore the spatially anisotropic triangular lattice and provide evidence that supports the existence of a gapped spin liquid between the square and triangular lattices.

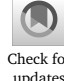

# 1 Introduction

The prototype of frustated quantum spin models is the spin-$\frac{1}{2}$ isotropic triangular lattice Heisenberg antiferromagnet (TLHAF). P. W. Anderson originally proposed that the ground state of the TLHAF was a resonant valence bond state [1], the precursor of the current notion of quantum spin liquids (QSLs) [2–4]. Now, it is known that the real ground state has a long-range magnetic order composed of three sublattices whose spins form angles of 120° with one another [5–10]. Nonetheless, the staggered magnetization has an important reduction from the classical value due to the quantum fluctuations, which are enhanced by frustration [8–10]. In fact, it has been proposed that the TLHAF lies close to a quantum melting point [11, 12], as it is sufficient to add a small next-nearest neighbor interaction of about $J_2/J_1 \simeq 0.07$ to drive the system to a QSL [13–28].

The proximity to a quantum critical point causes unusual behaviors in the dynamic properties [29]. For example, recent inelastic scattering experiments on $Ba_3CoSb_2O_9$, considered to be the experimental realization of the TLHAF (plus a small interlayer interaction $J' \sim 0.05J$ and XXZ easy-plane anisotropy $\Delta \sim 0.9$ [30–33]), show anomalous characteristics in the dynamical structure factor such as an extended continuum of excitations [32–34]. This feature cannot be explained by the most conventional theories for ordered magnets [12, 35, 36].

The new perovskite $Ba_8CoNb_6O_{24}$, which is formed from $Ba_3CoSb_2O_9$ by adding more nonmagnetic layers such that the interlayer interaction is reduced practically to zero, provides an even closer realization of the TLHAF [37, 38]. In fact, it was proposed that $Ba_8CoNb_6O_{24}$ provides a nearly ideal experimental illustration of textbook Mermin-Wagner physics in a two-dimensional magnetic system down to 0.1 K as it presents no sharp features down to that temperature, suggesting the absence of a finite-temperature magnetic phase transition. However, below that value, the system appears to break the trend towards zero-temperature magnetic order, seemingly going to a QSL. Furthermore, the specific heat $c_v$ measurements of this compound show a broad peak which is not well reproduced by the standard high-temperature series expansions (HTSE) calculations [7, 37–39]. Recent tensor product calculations on finite systems at finite temperature on triangular lattices find that aside from the main peak at $T_h \sim 0.55$ in the $c_v$, another peak or shoulder appears at lower temperatures $T_l \sim 0.2$, consistent qualitatively with the wide peak observed experimentally [39]; but a good theoretical description of the experimental measurements is still lacking.

In fact, the accurate description of the low-temperature properties of quantum spin systems presents a difficult challenge for theoretical and numerical methods in general. On one side, the numerical methods that depart from the purely quantum solution at $T = 0$, such as exact diagonalization (ED) and the tensor methods, struggle with lattice sizes and computation times as the temperature increases. On the other hand, quantum Monte Carlo works naturally at finite temperature but fails in the presence of magnetic frustration due to the infamous sign problem. Finally, the HTSE depart from the $T = \infty$ limit and can only get as low as $T \sim J$, whenever there is no finite-temperature phase transition [40].

Several interpolation methods have been proposed to overcome this limitation of the HTSE [41–46]. In particular, in the absence of finite-temperature phase transitions, the so-called

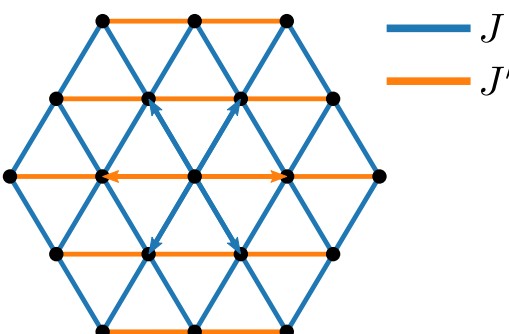

Figure 1: Spatially anisotropic triangular lattice, with $J$ exchange interaction running along two directions (purple) and $J'$ running along the remaining one (green).

"entropy method" (or HTSE+$s(e)$) has proven to obtain reliable results for the specific heat $c_v$ [47–51]. The main problem of the method is that it depends on the knowledge of the low-temperature behavior of the specific heat and the ground-state energy which is not usually available for quantum models. Nonetheless, looking for the highest number of coinciding Padé approximants has proven to be a way to narrow down the parameter space, allowing to obtain certain parameters self-consistently [44–46].

In this article, we use the entropy method to study the vicinity of the TLHAF. For the TLHAF, we show that there is a narrow region of ground-state energies with over 80% of coinciding Padé approximants. These energies are in perfect agreement with the available calculations from methods such as ED and density-matrix renormalization group (DMRG). We then calculate the thermodynamic quantities such as the specific heat and the entropy and compare them with the recent experimental measurements in the perovskite $Ba_8CoNb_6O_{24}$, finding a considerable improvement with respect to the raw HTSE calculations and other methods. Then, we analyze what kind of further interactions could improve the comparison with experiments. Finally, we explore the spatially anisotropic triangular lattice, where we find clear signals of quantum phase transitions at $T = 0$, and compare with existing results.

The article is organized as follows: in section 2, we present the spatially anisotropic TLHAF along with the known limits and the existing results. Section 3 recalls the main lines of the entropy method, that relies on the knowledge of the HTSE and the low-temperature properties of the system. Results are presented in section 4, first for the TLHAF and then including spatial anisotropy, and compared with experimental and theoretical results. Finally, concluding remarks and future perspectives are given in section 5.

## 2 Heisenberg model on a spatially anisotropic triangular lattice

The spatially anisotropic TLHAF is defined as,

$$\mathcal{H} = J \sum_{\langle ij \rangle} \mathbf{S}_i \cdot \mathbf{S}_j + J' \sum_{\langle ij \rangle'} \mathbf{S}_i \cdot \mathbf{S}_j, \tag{1}$$

where $J$ runs along two of the three triangular directions and $J'$ runs along the remaining one (see figure 1), and $\mathbf{S}_i$ are the quantum spin-$\frac{1}{2}$ operators. This model contains some well-known limits, such as the isotropic triangular lattice when $J = J'$. In this case, the ground-state energy has been determined to be approximately $e_0 = -0.544(2)$ by ED [8], DMRG [52], and variational Monte Carlo (VMC) [9]; and the staggered magnetization is $m = 0.205(15)$ from VMC [9] and DMRG [10] calculations (about 40% of the classical value [8]). Since the

ground state is antiferromagnetically ordered, the specific heat at low temperatures behaves as $c_v \propto T^2$.

On the other hand, when $J' = 0$ the model becomes the square lattice Heisenberg model. In this case the ground-state energy has been determined accurately with quantum Monte Carlo calculations, since there is no sign problem. Even though the results may vary from one author to another, most of them agree that $e_0 = -0.669(1)$ [53, 54]. The ground state corresponds to an antiferromagnetic Néel order with a higher staggered magnetization than in the triangular lattice, $m = 0.307(1)$ [54]. The specific heat at low temperatures also behaves as $c_v \propto T^2$.

When $J = 0$ the model becomes a set of decoupled spin-$\frac{1}{2}$ chains, and the solution is known exactly from the Bethe ansatz [55]. The ground-state energy is $e_0 = 0.25 - \ln 2$ and the ground state is a gapless QSL without long-range magnetic order, but with a quasi-long-range order and algebraically decaying correlations along the chain. In this case, the low-temperature specific heat behaves as $c_v \propto T$.

This model may be relevant in several compounds where the effective spins lie on chains coupled triangularly. Some of them are inorganic, such as $Cs_2CuCl_4$ and $Cs_2CuBr_4$ [56–59], and some are organic like $\kappa$-$(ET)_2Cu_2(CN)_3$ and $Me_3EtSb[Pd(dmit)_2]_2$ [60–64]. On the other hand, there are some theoretical studies, but none of them fully agree on the corresponding quantum phase diagram [13, 28, 64–68]. On one side, for $J'/J > 1$, some VMC studies predict a two-dimensional QSL phase for $1.18 \lesssim J'/J \lesssim 1.67$ and a one-dimensional one for the decoupled chains limit ($J'/J \gtrsim 1.67$) [66, 69]. However, a more recent VMC study argues that the spiral phases survive up to $J'/J \simeq 1.67$ [68]. For $J'/J < 1$, the model has been less studied and Schwinger bosons theory predict a gapped disordered phase for $0.6 \lesssim J'/J \lesssim 0.9$ [13, 28], while VMC calculations predict that if there is indeed a QSL, it can only be in the range $0.7 \lesssim J'/J \lesssim 0.8$ and it would be gapless [68].

# 3 Entropy Method

In this section we present the main aspects of the entropy method, an interpolation method that relies on the knowledge of the HTSE and some ground-state properties such as the ground-state energy and the leading term in the low-temperature specific heat [41–45]. The HTSE of the free energy per spin $f$ is defined as

$$\beta f = -\ln 2 - \sum_{i=1}^{n} c_i \beta^i + O(\beta^{n+1}), \qquad (2)$$

where $\beta = 1/T$ is the inverse temperature and $c_i$ are polynomials of $J$ and $J'$. For unsolved models, the series are only known up to some finite order $n$ and they only converge down to temperatures of around $J$. This range can be extended down to $J/2$ using Padé approximants [40].

From this series, we can determine the series of the entropy and the energy as a function of $\beta$, respectively $s(\beta)$ and $e(\beta)$. Then we obtain the coefficients of the series of $s(e)$ up to the same order $n$ by solving $s(e(\beta)) = s(\beta)$. This thermodynamic quantity $s(e)$ is bounded by $\ln 2$ at $e = 0$, corresponding to $T = \infty$, and 0 at the ground-state energy $e_0$, corresponding to $T = 0$. Since $\beta = s'(e)$, there is a singularity at $e_0$ given by $\beta \to \infty$. The nature of the singular behavior of $s$ close to $e_0$ can be deduced from the low-$T$ behavior of the specific heat $c_v(T)$.

In order to extend the series to the whole range of energies, the singular behavior has to be removed. When $c_v(T) \sim (T/T_0)^\alpha$ at low-$T$ this is done by constructing an auxiliary function,

$$G(e) = \frac{s(e)^\mu}{e - e_0}, \qquad (3)$$

where $\mu = 1 + 1/\alpha$. In two dimensions, $\alpha = 2$ for ordered antiferromagnets and $\alpha = 1$ for gapless disordered states or ferromagnets. If no finite-temperature transition occurs, $G(e)$ is a positive regular function of $e$ whose HTSE is deduced from that of $s(e)$. For this function, all Padé Approximants with poles or roots in the energy range $[e_0, 0]$ are non physical and discarded directly.

From the Padé approximants $G_{PA}(e)$ of $G(e)$ the entropy is evaluated for all energies in $[e_0, 0]$ as:

$$s_{PA}(e) = [G_{PA}(e)(e - e_0)]^{1/\mu} . \tag{4}$$

Note that only the physical Padé approximants are considered, the non-negative ones in the range $e_0 \leq e \leq 0$. Then, taking into account that $\beta(e) = s'(e)$ and $c_v(e) = -s'^2(e)/s''(e)$, we obtain $c_v(T)$ from $c_v(e)$ and $T(e) = 1/\beta(e)$. This construction depends on the choice of $\alpha$ and the ground-state energy $e_0$. However, it has been suggested that $e_0$ can be obtained self-consistently by maximizing the number of coinciding Padé approximants (CPAs). This criteria has recently proven to give good results in several cases [44–46].

The number of CPAs is determined by obtaining all the functions $c_v(T)$ from the physical Padé approximants of $G(e)$. Then, the one with the largest distance to their average is discarded. The procedure continues up until all the functions are within a certain threshold $\delta$ from their mean value. Unless otherwise specified, we use $\delta = 0.001$.

We also consider a low-temperature behavior as $c_v(T) \sim T^2 \exp(-T_0/T)$ corresponding to a gapped ground state, where $T_0$ is the gap. The auxiliary function is defined as

$$G(e) = -\left(\frac{s(e)}{e - e_0}\right)'(e - e_0), \tag{5}$$

and it requires an integration to recover $s_{PA}(e)$ [44]. We call this case $\alpha = 0$.

$T_0$ is related to $G(e_0)$ by

$$T_0 = \frac{\alpha + 1}{\alpha^{1+1/\alpha} G(e_0)}, \quad \text{or} \quad T_0 = \frac{1}{G(e_0)}, \tag{6}$$

for gapless and gapped systems, respectively.

# 4 Results

## 4.1 Isotropic triangular lattice

For this case ($J' = J$), we have computed the HTSE up to order $n = 18$ (previous calculations were at order $n = 13$ [41]). In this part we fix $\alpha = 2$ since we assume an antiferromagnetically ordered ground state.

To find the optimal ground-state energy at a given HTSE order, we perform sweeps in energy at steps of $10^{-5}$ in a wide range around the expected values and check for the proportion of CPAs (pCPA). This quantity is expressed with respect to the total number of Padé approximants ($n + 1$ at order $n$). In figure 2, we show the pCPA results using the series up to orders from 14 to 18 (top panel), and also using groups of 3 consecutive orders (bottom panel). Using three consecutive orders is a way to measure the convergence with $n$. If converged with $n$, pCPA from 3 consecutive orders will be close to the individual pCPA of a single order. Otherwise, it will be much smaller. The grey area in figure 2 represents the ground-state energy found with ED along with its uncertainty [8]. In figure 2, we see that the peaks in the pCPAs are thinner, and closer to the ED value, as the order increases. At the peaks, we get about 80% of CPAs (i.e. about 15 CPAs at the largest order, and about 43 CPAs using the three largest orders).

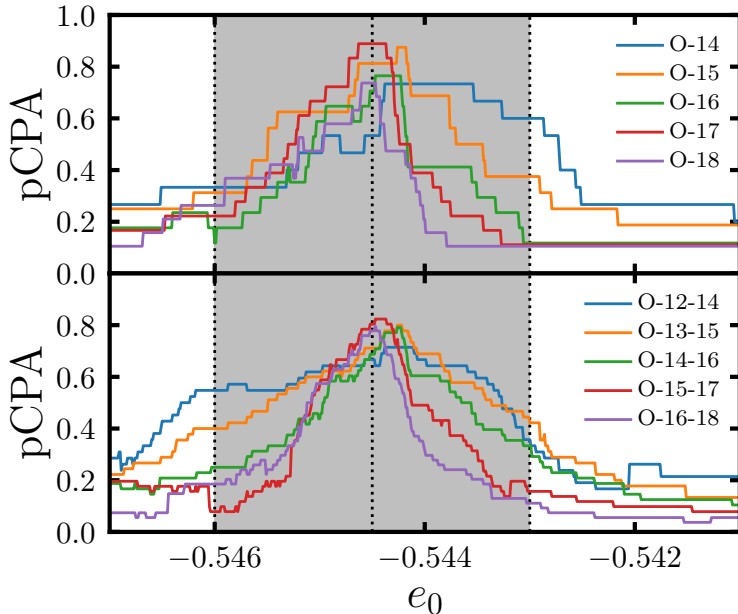

Figure 2: Proportion of coinciding Padé approximants (pCPA) as a function of the ground-state energy for several orders $n$ (top panel) and also taking groups of 3 orders (bottom panel). The grey area denotes the value from exact diagonalization along with its error bars, $e_0 = -0.5445(15)$ [8].

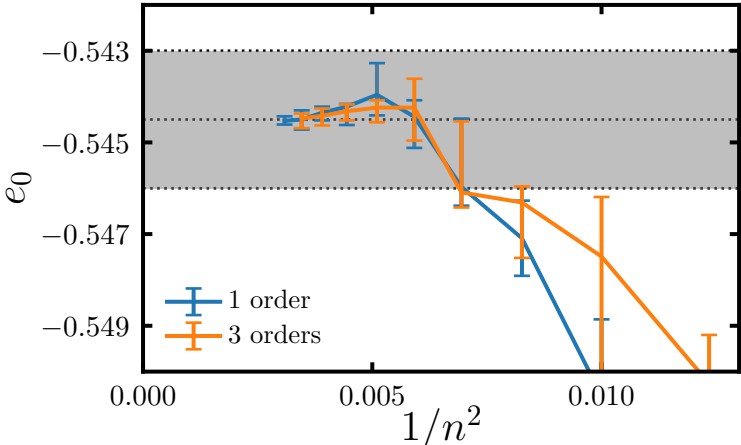

Figure 3: For the TLHAF: Best values of the ground-state energy per site $e_0$ (with corresponding uncertainties) for several orders $n$ (blue) and groups of 3 orders $[n-1, n+1]$ (orange), as a function of $1/n^2$. The grey area denotes the value from exact diagonalization (with uncertainties) [8].

Figure 3 shows the best values of $e_0$, extracted from figure 2, as a function of $1/n^2$. We tried several scaling types and found that this one was the best. However, we do not have any microscopic argument to support it. For the cases of 3 orders, $n$ is the mid-value. Uncertainties are evaluated as the width at $\text{pCPA}(e_0) - 0.1$. The grey area shows again the ED value [8]. We see that these estimations of $e_0$ vary smoothly for orders $n \geq 13$ and seem to extrapolate around $0.545(1)$. For order $n = 18$ we get $e_0 = -0.5445(1)$, and using orders $n = 18$ to $16$ we get $e_0 = -0.5445(2)$.

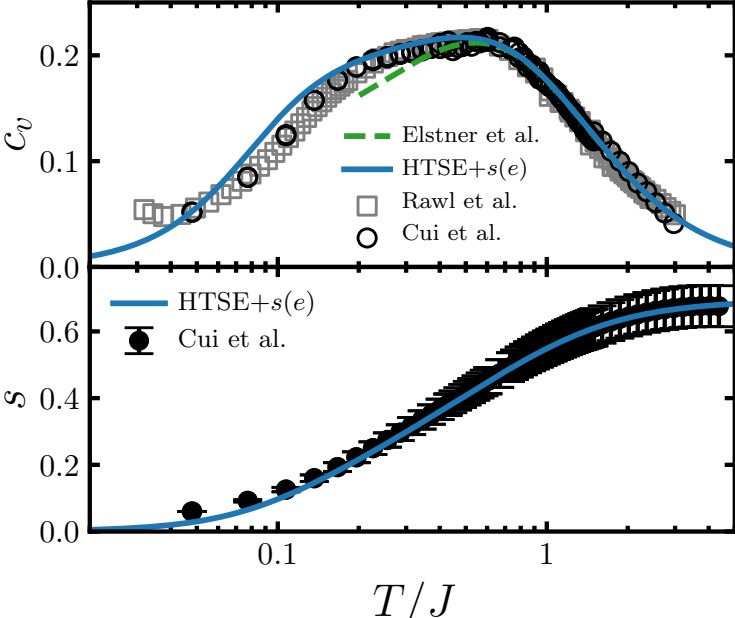

Figure 4: Calculations of the specific heat $c_v$ (top panel) and entropy $s$ (bottom panel) of the triangular lattice Heisenberg model using HTSE+$s(e)$, compared with the experimental measurements made by Rawl et al. and Cui et al. on the compound $Ba_8CoNb_6O_{24}$ [37,38], and with the Padé Approximants calculations by Elstner et al. [7].

All the values we have presented are close to those previously obtained with the same method [41]. Indeed: (a) in the previous calculations the ground-state energy was an input and not a calculated quantity, but we get values very similar to the ED ones which were used, and (b) we now have access to the HTSE at order $n = 18$ compared with the previous $n = 13$ calculations.

From the best value of $e_0$ at each order of the HTSE, we calculate the thermodynamic quantities such as the specific heat $c_v(T)$ and the entropy $s(T)$ at all temperatures. These quantities converge rapidly with the HTSE order $n$. For example, $c_v(T)$ shows a round peak at $n = 10$ that becomes wider and flatter at $n = 13$, and does not change appreciably up to order $n = 18$. In figure 4 we show our results using the HTSE+$s(e)$ at order $n = 18$ and compare them to the experimental results $Ba_8CoNb_6O_{24}$ [37,38]. The experimental value of the exchange energy is $J = 1.66 \pm 0.06$ K [38]. Here, the error introduced by the uncertainty in $J$ is smaller than the symbol size. In the case of $s(T)$, a vertical shift in the experimental data is mandatory in order to reach the high-$T$ limit of $\ln(2)$. Since $s$ is obtained experimentally by integration of $c_v/T$, for which there are no measurements down to $T = 0$, this shift results in a residual entropy of $s^* = 0.06$ at $T^* = 0.08$ [38,39].

In the top panel of figure 4 we see a good agreement between $c_v(T)$ obtained for the isotropic triangular Heisenberg model and the measurements on the triangular compound $Ba_8CoNb_6O_{24}$. In fact, the HTSE+$s(e)$ calculation represents the wide peak in $c_v$ much better than the raw HTSE calculations [7,38,39]. However, some features do not fully match with the experiments such as a smaller experimental $c_v$ in the temperature range $0.05 < T/J < 0.2$. These small differences at low $T$ could be related to the fact that $Ba_8CoNb_6O_{24}$ breaks the tendency to develop a long-range magnetic order at $T = 0$ and behaves as a QSL [38]. This behavior would imply that $c_v$ is no longer proportional to $T^2$ at very low $T$. Nevertheless, using $\alpha = 1$ or $\alpha = 0$ does not fit correctly the main peak.

Contrary to the latest computational results obtained with exponential tensor renormalization group [39], we do not find a double-peak structure in $c_v$ (also not present in the experimental results). However, the wide peak may originate in two temperature scales in the excitation spectra [39].

We find that, at $n = 18$ in the HTSE+$s(e)$, the peak of the specific heat is $c_v(T^*) = 0.2169(1)$ at $T^* = 0.4803(1)$, taking an average and twice the standard deviation from all the values corresponding to the 14 coinciding Padé approximants. The value of $T^*$ is lower than the one of the raw HTSE and the numerical results on finite lattices, $T_M = 0.55$ [39]. These values make sense since they are very close to the energy scale of the rotonic modes of the triangular lattice [12, 70], between $0.55J$ and $0.6J$, and most of the spectral weight lies below these values [12].

The low-$T$ energy scale is measured by $T_0 = 0.200(1)$ (see equation 6). This value can be compared with the one obtained using Schwinger bosons at the mean-field level $T_0 = 0.137$, and considering only the physical low energy excitations $T_0 = 0.192$ [71]. From the spin-wave velocity in the Debye construction it is $T_0 = 0.294$ [71].

Finally, in the bottom panel of figure 4, we see that the entropy calculations match very well the experimental measurements, showing only small discrepancies at the lowest temperature. It is worthy mentioning that our method captures both the high-$T$ limit and the low-temperature behavior of the entropy for the TLHAF.

## 4.2 Square lattice limit and isolated chain limit

As mentioned above, there are two important limiting cases when we take into account spatial anisotropy ($J \neq J'$). On one side we have the square lattice limit when $J' = 0$ (we take $J = 1$), and on the other side we have the one-dimensional chain limit when $J = 0$ (and $J' = 1$). Exact results from QMC for the square lattice and the Bethe ansatz for the single chain allow to benchmark the entropy method for these limit cases. These cases were also treated with this method in the original article [41], finding a good number of CPAs when imposing the energy as the exact one. Here, $e_0$ is obtained self-consistently as explained above. Also, we use higher orders in the HTSE: $n = 20$ for the square lattice (compared with $n = 14$ [41]), and $n = 28$ for the single chain (compared $n = 24$ [41]).

Figure 5 shows the best ground-state energies $e_0$ for several orders, both taking one by one and in groups of three (same as in the isotropic triangular lattice). In the top panel we show the results for the single chain with $\alpha = 1$, where the exact result from the Bethe ansatz $e_0 = 0.25 - \ln(2) \simeq -0.443147$ is shown in a dotted line [55]. The results obtained with the entropy method are very close to this exact value, even for orders as low as $n = 14$ for which we get $e_0 = -0.44277(1)$. At order $n = 28$ we get $e_0 = -0.4434(2)$ which lies close to the exact result, while using the three highest orders we get $e_0 = -0.44338(5)$. Regarding the pCPAs, the numbers are between 0.55 and 0.65.

The bottom panel of figure 5 shows the ground-state energy results for the square lattice. The dotted lines represent the quantum Monte Carlo results along with uncertainties: $e_0 = -0.6694(1)$ [54, 72]. At order $n = 20$, we get $e_0 = -0.666(1)$ and using the last 3 orders $e_0 = -0.6664(7)$, which are very close to the QMC result. The number of pCPAs lies between 0.5 and 0.6. Even though we still find more than half of the PAs that match for this lattice, the number is not as high as in the triangular lattice case where we had pCPA∼ 0.8.

## 4.3 Anisotropic triangular lattice

Now that we have tested the limiting known cases and confirmed that the entropy method gives good results in these three points (square, triangular, and chain), we explore the intermediate values of anisotropy. Keeping $J = 1$, we vary $J'$ between 0 (square limit) and 2.0

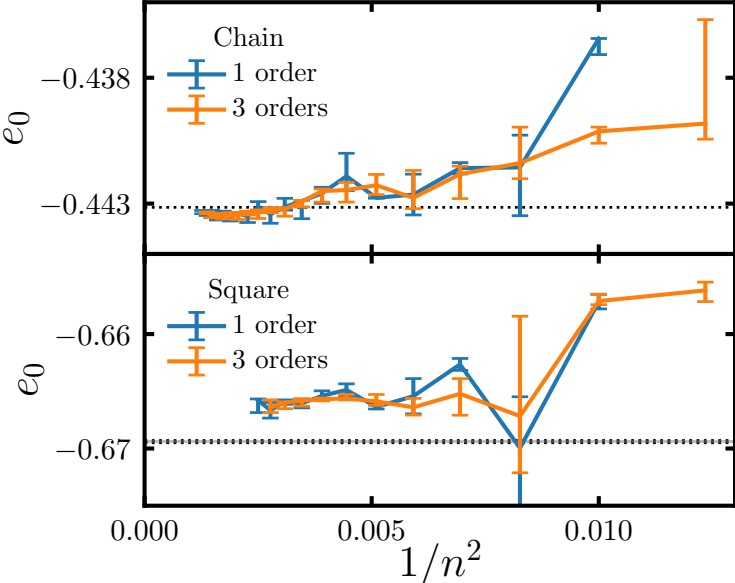

Figure 5: Best values of the ground-state energy $e_0$ (with corresponding uncertainties) for several orders $n$ (blue) and groups of 3 orders $[n-1, n+1]$ (orange), as a function of $1/n^2$. On the top panel: the chain limit, and on the bottom panel: the square lattice. The dotted lines stand for the exact result for the chain [55], and the quantum Monte Carlo result for the square lattice [54, 72].

(to approach the chain limit), passing through $J' = 1$ which is the triangular case. For the general model we have the series up to order $n = 16$ (compared with $n = 10$ and $n = 12$ for the magnetic susceptibility [58, 73]). In this section we use the values $\alpha = 0$, 1, and 2 in order to represent different possible phases (gapped spin liquid, gapless spin liquid, and antiferromagnetic order). However, there is no variational principle in the method and we cannot select the phase by looking at the lowest energy. Instead, we use the pCPAs along with the known limits and the continuity of $e_0$ to propose phase transitions.

### 4.3.1 From the square to the triangular lattice ($0 \leq J' \leq 1$)

In figure 6-(a) we show the results for the best ground-state energies along with the corresponding uncertainties for the cases where $\alpha = 0$ (gapped QSL), 1 (gapless QSL), and 2 (long-range order). We know that at both of the extremes ($J' = 0$ and 1) the correct solution corresponds to $\alpha = 2$, which agrees with the black circle and square from QMC and ED [8, 54], respectively. However, for intermediate values of $J'$ there is no consensus in the literature regarding the type of ground state. Nonetheless, $e_0(J')$ being a continuous function, solutions with different $\alpha$ can only occur at crossing points.

For $J' < 0.6$, the functions $e_{0,\alpha}(J')$ are clearly different. We can assume that the magnetic Néel order stays up to $J' = 0.6$, in agreement with the VMC [68] and Schwinger bosons theory (SBT) results [28].

Close to the triangular lattice the energies of the solutions $\alpha = 0$ and $\alpha = 2$ are very similar, allowing the possibility of a phase transition for any value $J' > 0.8$. The remaining solution $\alpha = 1$ corresponding to a gapless QSL always presents a higher ground-state energy than the other two, except for values around $J' \simeq 0.7$. However, in the range $0.6 \leq J' \leq 0.8$ pCPA presents a dip for each $\alpha$ (see figure 6-(b)), meaning that the solutions cannot be trusted. The failure of the entropy method in this region can be interpreted in many ways. On one side, the

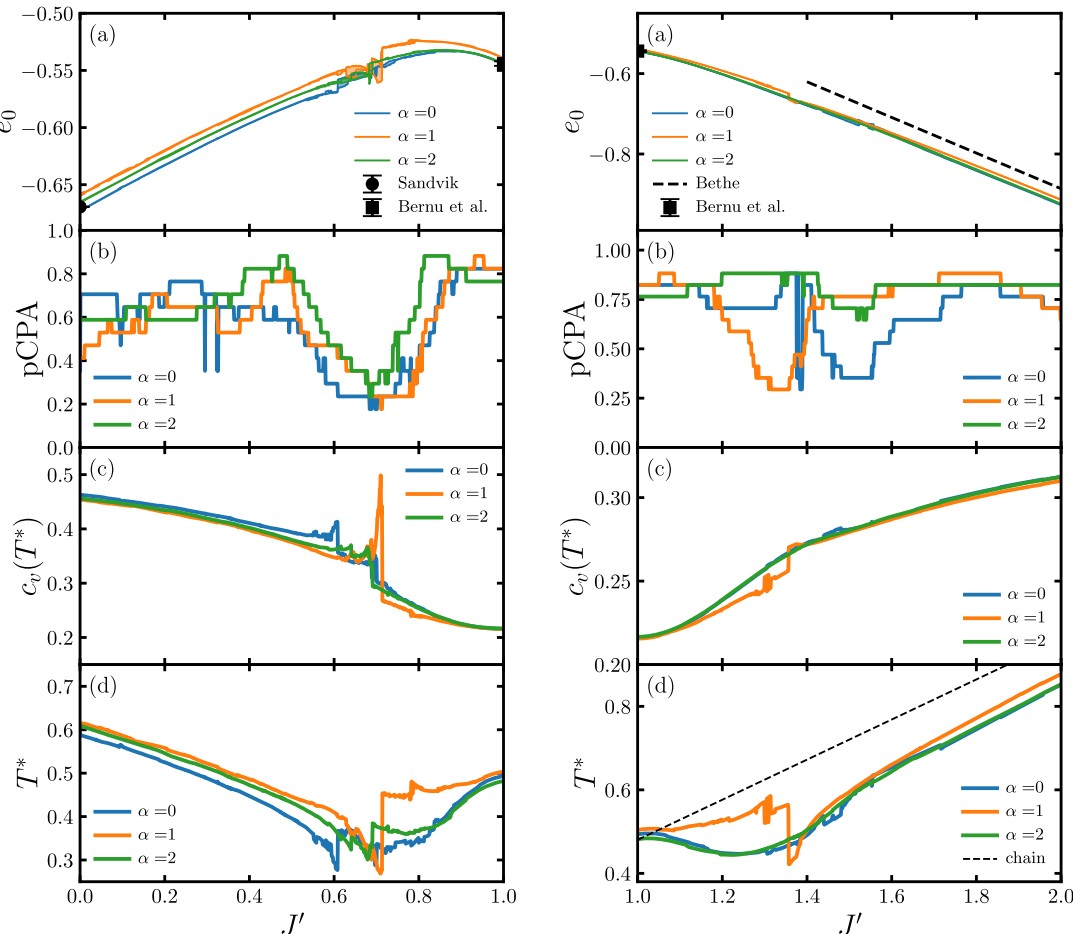

Figure 6: Entropy method results as a function of $J'$ using $n = 16$ for $\alpha = 0$, 1, and 2 (in blue, orange, and green, respectively). On the left panels, between the square ($J' = 0$) and triangular lattices ($J' = 1$); and on the right panels between the triangular lattice and $J' = 2$. Panels show: (a) the ground-state energy with corresponding uncertainties (black symbols correspond to ED [8] and QMC [54] and the dashed line corresponds to the exact energy of a single chain [55]); (b) the proportion of coinciding PAs; (c) the value of the peak in the specific heat; (d) the temperature at which the peak appears (In black dashed lines are the results for the single chain with $\alpha = 1$ at order 20).

basic assumption of this interpolation method is the absence of singularities in the thermodynamic functions, something which is not true if a finite-temperature transition exists. Since in two dimensions the Mermin-Wagner theorem precludes the possibility of any continuous symmetry breaking at finite temperature [74], the transition can only occur due to a discrete symmetry breaking of an effective order parameter. This is believed to happen in systems such as the $J_1 - J_2$ model on the square lattice where an emergent Ising order parameter induces a finite-temperature phase transition [75–77]. Another example is the classical Heisenberg model on the $J_1 - J_3$ kagome lattice, where a Potts transition occurs [45]. For this last model with spin-$\frac{1}{2}$, the entropy method presents a dip in pCPAs around the classical transition [45]. However, for the present model, such a transition has not been found so far. Another possibility is that the entropy method fails because the thermodynamics involve the contribution of different states and the HTSE at order 16 does not capture such physics. At least, the method

is able to detect something unusual occurs.

If no finite-temperature phase transition occurs, the thermodynamic functions should change smoothly in this region with respect to $J'$. This is also true for the height of the peak of $c_v(T^*)$, but not necessary true for the position $T^*$ (see figure 6-(c) and ((d)). A crossing in all the quantities ($e_0$, $c_v(T^*)$, and $T^*$) suggests a transition between $\alpha = 2$ and $\alpha = 0$ at $J' \simeq 0.6$, and again between 0.8 and 0.9. This region is similar to the region $0.7 \leq J' \leq 0.8$ for which the VMC calculations propose a close competition between an ordered state and a gapless QSL [68]. However, in our case the region is larger and the QSL would be of the gapped nature. These two features are in better agreement with the SBT calculations that predict a gapped QSL in the range $0.6 \leq J' \leq 0.9$ [28].

Figure 6-(c) shows that the peak of $c_v$ is the lowest in the triangular lattice limit, but, for $J' \simeq 0.7$ the position of the peak is at lower temperatures (see figure 6-(d)), indicating a larger density of states at low energy, something which agrees with the presence of a QSL in the region.

### 4.3.2  From the triangular lattice to the chain ($J' \geq 1$)

In figure 6-(a), we show the best ground-state energies obtained for each value of $\alpha$, with the corresponding uncertainties. In this case, the correct solution is represented by $\alpha = 2$ on the left side at the triangular limit, and by $\alpha = 1$ (gapless QSL) in the single-chain limit at $J' \to \infty$. However, at large values of $J'$ there is already a separation between the $\alpha = 1$ solution and the other two. At least one transition between both limits has to occur for intermediate values. Decreasing $J'$ from the one-dimensional limit, the energy of the $\alpha = 1$ solution crosses the other two solutions at about $J' \simeq 1.4$, where the gapless QSL solution has a low quality (see figure 6-(b) around $J' \simeq 1.3$).

The energy crossing is consistent with the crossing in the height and position of the peak in the specific heat that can be seen in figure 6-(c) and (d). However, we cannot determine if the transition is directly $\alpha = 2 \leftrightarrow 1$ or if there is another intermediate gapped phase $\alpha = 0$ since all the parameters corresponding to $\alpha = 2$ and 0 are very close in the range $1 \leq J' \leq 1.4$.

In figure 6-(d) we have also added $T^*$ for the single chain in black dashed lines, which should be reached at $J' \to \infty$. Even if the values at intermediate $J'$ are far from those of the single chain, we can see that the position of the peak $T^*$ is moving linearly with $J'$ above $J' \simeq 1.4$ for every value of $\alpha$, pointing out that $J'$ is the only relevant energy scale.

To sum up, we find that the system behaves as a one-dimensional spin liquid above $J' \simeq 1.4$. On the other hand, it is hard to ensure that the two-dimensional phase below $J' \simeq 1.4$ is magnetically ordered as determined by the VMC calculations which predicts a transition at $J' \simeq 1.7$ [68]; in which case the phenomenon would be the same as for the $S = 1$ system [78] regardless of the topological difference between the one-dimensional limits [79]. We cannot rule out the existence of a gapped QSL between the one-dimensional phase and the magnetically ordered one in the triangular limit. For $J \geq 1$, the picture agrees with previous VMC calculations predicting two transitions at $J' \simeq 1.2$ and $J' \simeq 1.7$ [66] and the SBT results predicting them at $J' \simeq 1.3$ and $J' \simeq 2.2$ [28]. It is clear that further studies are needed in order to elucidate the nature of this complicated phase diagram, since DMRG studies on the Hubbard model show a lot of different competing phases at large $U$ [80].

### 4.4  Comparison with $Ba_8CoNb_6O_{24}$

In this section we explore what kind of interactions could play a role in the specific heat of the triangular compound $Ba_8CoNb_6O_{24}$, whose measurements suggest that it is the experimental realization of the TLHAF [37,38]. However, the low-temperature behavior cannot be explained by the standard HTSE or any other method. To improve the comparison with experiments, we

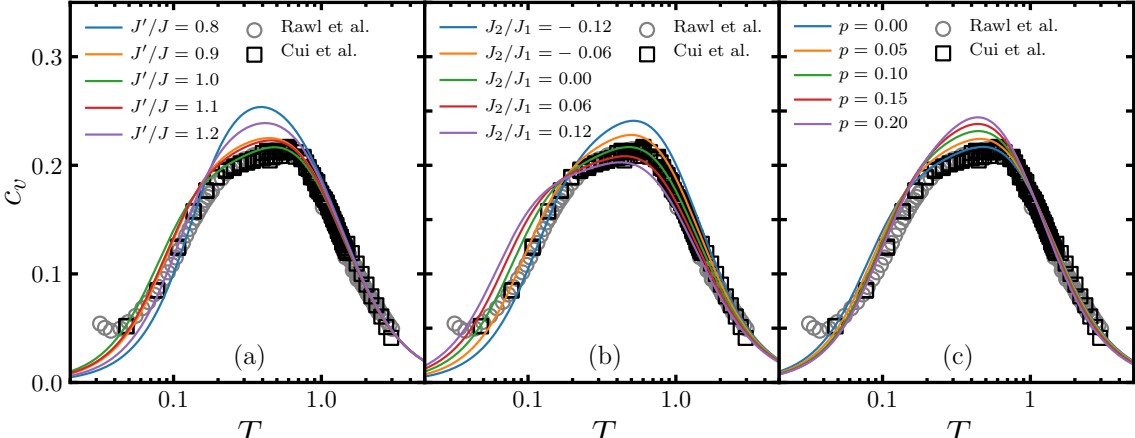

Figure 7: Specific heat $c_v$ for $\alpha = 2$ in several cases: (a) with spatial anisotropy, (b) with next-nearest neighbor interactions, and (c) with impurities. Experimental results are denoted by circles [37] and squares [38].

depart from the isotropic triangular lattice and add further interactions or anisotropies. The model is normalized to keep fixed the dominant term of $c_v(T)$ at high temperatures.

The first case is the spatial anisotropy introduced in the previous section, with $J'^2 + 2J^2 = 3$. In figure 7-(a) we show $c_v(T)$ for various values of $J'/J$ around the isotropic point. The peak is the lowest at the isotropic point, and the best agreement with experimental measurements are for $J'/J = 1.0$ (green line). Any degree of anisotropy makes the peak rounder and higher.

Next, we include next-nearest neighbor interactions:

$$\mathcal{H} = J_1 \sum_{\langle ij \rangle} \mathbf{S}_i \cdot \mathbf{S}_j + J_2 \sum_{\langle\langle ij \rangle\rangle} \mathbf{S}_i \cdot \mathbf{S}_j, \tag{7}$$

with $J_1^2 + J_2^2 = 1$. For this case we have access to the series up to order $n = 13$. Figure 7-(b) shows $c_v(T)$ for this model. Both at high and low temperatures, the experimental results are better reproduced with $J_2/J_1 \simeq 0$. For negative values of $J_2$, the magnetic order of the triangular lattice is enhanced since the interaction is non-frustrating, and the peak gets rounder and higher. For positive values of $J_2$ the interaction is frustrating and the peak gets flatter and wider.

Then we investigate the possibility of spin anisotropy. In this case the Hamiltonian reads

$$\mathcal{H} = J_{xy} \sum_{\langle ij \rangle} \mathbf{S}_i^\perp \cdot \mathbf{S}_j^\perp + J_z \sum_{\langle ij \rangle} S_i^z S_j^z, \tag{8}$$

where $\mathbf{S}_i^\perp = (S_i^x, S_i^y)$, with $2J_{xy}^2 + J_z^2 = 3$. We do not show the corresponding $c_v(T)$ calculations since they show that any value of spin anisotropy could reproduce well the experimental results, since there is no much change in $c_v$. However, it is expected that the spin susceptibility $\chi$ presents a large change with different values of $J_{xy}/J_z$, as it happens in the kagome antiferromagnet [44]. Thus, the specific heat does not seem a useful quantity to determine the degree of spin anisotropy present in a given compound. It is worth mentioning that the HTSE results used in ref. [37] to compare with experiments show a large change with respect to the spin anisotropy because of the change in the energy scale, which we keep constant by enforcing $2J_{xy}^2 + J_z^2 = 3$.

Finally, we explore the possibility of non-magnetic impurities. In this case we have access to the series up to the same order as in the triangular lattice, $n = 18$. We show in figure 7-(c)

the $c_v(T)$ calculations for several densities of impurities, where $p$ is the ratio of missing spins and goes from 0 to 0.2. The peak gets rounder and higher as $p$ increases, so that the best description of the experimental measurements is realized by the pure triangular case without impurities.

## 5 Conclusion

We have used the entropy method to study several ground-state and thermodynamic properties of different antiferromagnetic Heisenberg models on triangular lattices. The entropy method is an interpolation method based on the knowledge of HTSE, the ground-state energy and the nature of the ground state (i.e. if it is magnetically ordered or a QSL, and the dimensional behavior). Since it is difficult to have exact and accurate values of the ground state energies for generic spin models, we have explored the possibility of determining it self-consistently by looking for the highest number of coinciding PAs. We tested this on the triangular and square lattice, as well as on the single chain, finding good agreement with previous results in the literature. For the triangular lattice we find $e_0 = -0.5445(2)$ with over 80% of coinciding PAs; for the square lattice we find $e_0 = -0.666(1)$ with over 50% of coinciding PAs; and for the single chain we find $e_0 = -0.4434(2)$ with over 60% of coinciding PAs.

We then explored the spatially anisotropic triangular lattice, for which the two limiting cases (square and single chain) and the isotropic case are known. We find that over most of the anisotropy range, the entropy method provides reliable ground-state energies. We have shown that phase transitions can be detected by this method and we provided evidence that supports the existence of a gapped QSL for $0.6 \geq J'/J \geq 0.85$. This is in good agreement with the SBT calculations [28] but not with the VMC ones [68]. On the other hand, between the triangular lattice and the single-chain limit, we are able to find indices of the one-dimensionalization phenomenon, but we cannot determine whether there is or not an intermediate two-dimensional spin liquid phase between both limits. A question which still remains open in the literature.

Finally we compare with the experimental specific heat and entropy on $Ba_8CoNb_6O_{24}$, which is considered to be the experimental realization of the TLHAF except for very low temperatures [37, 38]. Indeed, we find that the TLHAF describes quantitatively well the experimental specific heat $c_v$. We rule out the existence of spatial anisotropy or further neighbor interactions, even though the combined effect of both interactions could be studied. Also, we show that the specific heat does not allow to determine the degree of spin anisotropy and one should rely on the magnetic susceptibility to measure it [44]. Only at temperatures of about $T \lesssim 0.1$ our calculations depart from the experimental results, showing that $Ba_8CoNb_6O_{24}$ is breaking the perfect triangular lattice Heisenberg model tendency below this temperature. The question of what happens still remains open.

## Acknowledgements

The authors would like to thank Y. Cui, W. Yu, and H. D. Zhou for sharing their experimental results, and C. D. Batista for the fruitful discussions.

**Funding information** This work was supported by the French Agence Nationale de la Recherche under Grant No. ANR-18-CE30-0022-04 LINK and by an *Emergence(s)* project funded by the Paris city.

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
