# Peer review of "Ground-state and thermodynamic properties of the spin-$\frac{1}{2}$ Heisenberg model on the anisotropic triangular lattice"

_SciPost Physics, doi:SciPost Phys. 12, 112 (2022)_

## Round 1 · Referee Report · Anonymous (Referee 1) · 2022-1-5

Report

The article entiteled "Ground-state and thermodynamic properties of the spin-1/2 Heisenberg model on the triangular lattice" by Gonzalez and collaborators uses high-temperature series expansions in combination with the entropy method, which is a specific extrapolation tool, to calculate and analyze the ground-state energy and the specific heat of the Heisenberg model on the anisotropic triangular lattice. The latter includes as limiting cases the 1d chain, the square lattice, and the isotropic triangular lattice, which are all well known from previous analytical and numerical studies. In addition, extensions of the model including spatial anisotropies, next-nearest neighbor Heisenberg interactions, and non-magnetic impurities are also considered which are in particular motivated by experimental findings on the frustrated quantum magnet Ba_8CoNb_6O_24. In my opinion the article is well written and the topic is interesting. Globally, I therefore recommend publication in SciPost. I nevertheless have some points, which the authors should address to further improve their manuscript.

Requested changes

List of points:

  • Title: I wondered whether one should put "anisotropic triangular lattice" in the title which expresses better the focus of the article
  • Figure 3, page 7: Is there a microscopic reason why a "1/n^2" scaling is used? If yes, maybe one can specify the argument. If not, one should mention it.
  • Page 8: There are two equations in the text which are too wide.
  • Page 9/10: Is there are a (physical) reason why the quality of the extrapolation seems worse on the square lattice compared to the triangular and chain limit? Naively I would have expected that the square lattice is the simplest system.
  • Page 13: Is there a simple argument why the specific heat does not (almost does not) depend on the spin anisotropy?
  • Page 13: Another possibility/option could be the presence of multi-spin interactions. Can the relevance of such interactions be excluded for the considered material?
  • Page 15: Title of reference [18] seems to be spoiled.

  • validity: top
  • significance: high
  • originality: good
  • clarity: high
  • formatting: good
  • grammar: good

Author:  Matías Gonzalez  on 2022-02-14  [id 2200]

(in reply to Report 1 on 2022-01-05)

We provide the response to the report in the attached file.

Attachment:

response1.pdf

---

## Round 1 · Referee Report · Anonymous (Referee 2) · 2022-1-17

Strengths

See report

Weaknesses

See report

Report

The present work deals with applications of the entropy method to calculate the
ground state energy and specific heat c(T) for spatially anisotropic triangular spin
lattices. The results are also compared with measurements [37,38] on the perovskite
Ba8CoNb6O24, which is considered to be a good realization of the isotropic triangular
lattice.
The entropy method [41] interpolates between the high-temperature behavior (given
by the power series expansion in the inverse temperature) and the low-temperature
behavior of the specific heat, which must usually be assumed to be known. However,
the latter is often a problem if, for example, the ground state energy is only known
imprecisely. In this situation, the authors offer a captivating solution approach, which
has also been used in previous work [44-46], but is the focus of the present paper:
The ground state energy chosen is the one for which the largest majority of consistent
Padé approximants of different order (m,n) for c(T) can be found. This approach is
compared with the known values for certain limiting cases (isotropic triangular
lattice, square lattice, system of disconnected chains) and proves to be surprisingly
accurate.
After calculating c(T) based on these results, a comparison with the experimental data
yields the somewhat sobering result that the interpolation does not provide a
significant improvement over the usual Padé approximations published in [7,37,38].
These approximations already reproduce the broad maximum of c(T), which agrees
with the experiments. For low temperatures the interpolated c(T) correctly goes to 0,
as expected based on the approach, but deviates from the experimental results.
Analogous limitations arise for the spatially anisotropic lattice in the transition ranges
between J=J'=1 and J'→0 on the one hand and J'→ꝏ on the other: in the intermediate
range, the present method fails at certain points for reasons that do not become
entirely clear, and one can only say, "At least the method is able to detect something
unusual."
Nevertheless, this is a beautiful work that seems worthy of publication in SciPost
Physics. After all, it is questionable to appreciate only positive results. The
presentation of the method and the discussion of the state of research and the physical
significance of the results are very well done. Before the final decision, I still
recommend the following minor points to be considered in a possibly revised
manuscript:

Requested changes

  1. In the Introduction it is not quite clear what TLHAF means: the general anisotropic or the isotropic triangular lattice. It should also be made clear that Ba8CoNb6O24 is always understood as an isotropic triangular lattice, notwithstanding the remarks in section 4.4.
  2. The last sentence before (5) “When the ground state is a gapped spin liquid, we have c_ν(T)~T^2 exp(-T_0/T)...” is insofar misleading as the restriction to α=2 of the general power term T^α is only due to the entropy method, not to the physics.
  3. In Section 4.1, one would like to find reasons why groups of 3 orders are formed for the pCPA. Obviously, this leads to a smoothing and to an improvement of the approximation. Why?
  4. Padé approximants are rational functions in T which may have poles on the positive T axis. Usually, these cases will have to be discarded (but see Figure 3 in [7]). This problem is not mentioned in the paper. Do no poles occur, or are these cases discarded and simply do not belong to the majority of consistent approximants?

  • validity: -
  • significance: -
  • originality: -
  • clarity: -
  • formatting: -
  • grammar: -

Author:  Matías Gonzalez  on 2022-02-07  [id 2168]

(in reply to Report 2 on 2022-01-17)

We provide the response to the report in the attached file.

Attachment:

response2.pdf

Anonymous on 2022-02-09  [id 2178]

(in reply to Matías Gonzalez on 2022-02-07 [id 2168])

Based on the authors' responses and the changes made, I have no objection to publication of this article, which represents a valuable contribution to a current area of research.

Author:  Matías Gonzalez  on 2022-02-07  [id 2167]

(in reply to Report 2 on 2022-01-17)

We provide the response to the report in the attached file.

Attachment:

response1.pdf

Anonymous on 2022-02-14  [id 2201]

(in reply to Matías Gonzalez on 2022-02-07 [id 2167])
Category:
remark

This is the response to Report 1 and has been uploaded a second time in response to that report.

---

## Round 2 · Author Response

We thank both referees for positively taking into consideration the relevance of our work, as well as for their suggestions. Below we address all of their suggestions and questions, hoping that the revised manuscript will be suitable for publication in SciPost Physics.
Yours sincerely,
The authors

---

## Round 2 · List of Changes

• Title updated to “Ground-state and thermodynamic properties of the spin-$\frac{1}{2}$ Heisenberg model on the anisotropic triangular lattice”

  • We added the following sentence to Page 6: “We tried several scaling types and found that this one was the best. However, we do not have any microscopic argument to support it.”

  • Added “For the TLHAF” to the caption of Fig. 3

  • Fixed Ref. [18] and non-breaking inline formulas

  • We have added the curves from Ref. [7] to our Figure 4

  • Clarified the use of TLHAF in the Introduction

  • We have changed the prelude of equation (5) to: “We also consider a gapped ground state with a low-temperature behavior as $c_ν(T)\sim T^2 exp(-T_0/T)$, where T_0 is the gap.”

  • We have added the following explanation to Page 6: “Using three consecutive orders is a way to measure the convergence with $n$. If converged with $n$, pCPA from 3 consecutive orders will be close to the individual pCPA of a single order. Otherwise, it will be much smaller.”

  • We have clarified the treatment of the singularities in the Padé Approximants at the end of Page 4 by adding: “For this function, all Padé Approximants with poles or roots in the energy range $[e_0, 0]$ are non physical and discarded directly.”

---

## Editorial Decision

published